# Androgens In Men Study (AIMS): protocol for meta-analyses of individual participant data investigating associations of androgens with health outcomes in men

Bu Beng Yeap,[1,2] Ross James Marriott [iD] ,[3] Robert J Adams,[4] Leen Antonio,[5] Christie M Ballantyne,[6] Shalender Bhasin,[7] Peggy M Cawthon,[8] David John Couper,[9] Adrian S Dobs,[10] Leon Flicker,[11] Magnus Karlsson,[12] Sean A Martin,[13] Alvin M Matsumoto,[14,15] Dan Mellström,[16] Paul E Norman,[1] Claes Ohlsson,[16] Eric S Orwoll,[17] Terence W O'Neill,[18,19] Molly M Shores,[20,21] Thomas G Travison,[7,22] Dirk Vanderschueren,[23] Gary A Wittert,[13] Frederick C W Wu,[24] Kevin Murray [iD] [3]

For numbered affiliations see end of article.

**Correspondence to**
Dr Bu Beng Yeap;
bu.yeap@uwa.edu.au

## ABSTRACT

**Introduction** This study aims to clarify the role(s) of endogenous sex hormones to influence health outcomes in men, specifically to define the associations of plasma testosterone with incidence of cardiovascular events, cancer, dementia and mortality risk, and to identify factors predicting testosterone concentrations. Data will be accrued from at least three Australian, two European and four North American population-based cohorts involving approximately 20 000 men.

**Methods and analysis** Eligible studies include prospective cohort studies with baseline testosterone concentrations measured using mass spectrometry and 5 years of follow-up data on incident cardiovascular events, mortality, cancer diagnoses or deaths, new-onset dementia or decline in cognitive function recorded. Data for men, who were not taking androgens or drugs suppressing testosterone production, metabolism or action; and had no prior orchidectomy, are eligible. Systematic literature searches were conducted from 14 June 2019 to 31 December 2019, with no date range set for searches. Aggregate level data will be sought where individual participant data (IPD) are not available. One-stage IPD random-effects meta-analyses will be performed, using linear mixed models, generalised linear mixed models and either stratified or frailty-augmented Cox regression models. Heterogeneity in estimates from different studies will be quantified and bias investigated using funnel plots. Effect size estimates will be presented in forest plots and non-negligible heterogeneity and bias investigated using subgroup or meta-regression analyses.

**Ethics and dissemination** Ethics approvals obtained for each of the participating cohorts state that participants have consented to have their data collected and used for research purposes. The Androgens In Men Study has been assessed as exempt from ethics review by the Human Ethics office at the University of Western Australia (file reference number RA/4/20/5014). Each of the component studies had obtained ethics approvals; please refer to respective component studies for details. Research findings will be disseminated to the scientific and broader community via the publication of four research articles, with each involving a separate set of IPD meta-analyses (articles will investigate different, distinct outcomes), at scientific conferences and meetings of relevant professional societies. Collaborating cohort studies will disseminate findings to study participants and local communities.

**PROSPERO registration number** CRD42019139668.

## Strengths and limitations of this study

► The individual participant data (IPD) meta-analyses are likely to have higher statistical power and provide greater scope to control for important confounders and risk factors than previous meta-analyses on this topic.

► Investigators from nine large prospective cohort studies (each with n>1000 participants) have agreed to collaborate with additional studies to be identified from systematic review.

► Harmonisation will be required for some variables (eg, physical activity, alcohol consumption) that are recorded differently among the component studies, and aggregate-level data will be sought where IPD-level data are not available.

► As this is an observational study, it will not fully eliminate the possibility of confounding influences of unadjusted effects.

► However, unlike a randomised controlled trial, this study will provide a more comprehensive characterisation of temporal relationships between baseline androgen concentrations and health outcomes in community-dwelling adult males.

## INTRODUCTION

As men grow older, testosterone production and circulating concentrations of testosterone decline while comorbidities accumulate.[1–4] Older men, even those in very good health, have lower circulating testosterone concentrations compared with healthy young men.[5 6] Although results have been inconsistent, an increasing number of studies have reported associations of low endogenous testosterone concentration with poorer health outcomes, especially in older men. For instance, studies that have used liquid chromatography tandem mass spectrometry, widely regarded as the reference method for the measurement of total testosterone concentrations,[7] have reported associations of lower endogenous testosterone concentrations with (1) cardiovascular disease and all-cause mortality in some cases[8–15] but not others[8 16–20]; (2) some cancers but not others[21 22] and (3) dementia[23 24] but not laboratory measures of cognitive function.[25] Therefore, it remains unclear whether testosterone is a biomarker of ill health or a causal factor for diseases of ageing.

Currently, testosterone treatment is recommended for men who have symptoms and signs of androgen deficiency and low testosterone concentrations, due to disease of the hypothalamus, pituitary or testes (organic or pathological hypogonadism).[26–29] Randomised controlled trials of testosterone treatment in men aged 65 and older with low-normal testosterone concentrations without organic hypogonadism have shown modest benefits on sexual function, anaemia, self-reported physical function and mobility and volumetric bone density, but not on some objective measures of cognition over 12–36 months.[30–34] The effect of testosterone on major adverse cardiovascular events (MACE) remains unclear.[35 36] However, the selection criteria of these trials were such that the screening to enrolment ratio was 65:1, a highly selected population of older men.[32 33 36] Importantly, the trials were neither large enough nor long enough to determine the effects of testosterone on MACE, development of dementia, bone fractures and mortality.[36] Therefore, a meta-analysis of data from prospective cohort studies, with extended follow-up periods, provides opportunities for better understanding of the temporal profiles of the postulated associations between endogenous testosterone concentration and incident health outcomes. Meta-analyses of individual participant data (IPD) are generally preferable to meta-analyses of aggregate data (AD) in that they typically have higher statistical power and provide scope to control for important confounders and risk factors.[37 38] Furthermore, one-stage IPD meta-analyses are preferable to two-stage approaches because the former uses an exact likelihood to directly model the distribution of IPD, offers the convenience of using a standard set of diagnostic tools to assess model fit and can arguably provide greater flexibility, in terms of options for statistical modelling.[39 40]

The Androgens In Men Study (AIMS), an international collaboration of prospective cohort studies, will examine the associations of sex hormones (comprising testosterone and dihydrotestosterone as the major androgens, and oestradiol as the major oestrogen, in the circulation) with health outcomes that are major sources of morbidity and mortality in middle-aged and older men. The group will perform a series of IPD meta-analyses to clarify the influence of sex hormone exposures on major health outcomes, including heart attack, stroke and cardiovascular deaths, cancer, new-onset dementia and all-cause mortality, as well as provide information on social, demographic and behavioural factors that are associated with endogenous testosterone concentrations. This work will characterise robustly the associations of several sex hormones with health outcomes in men in general over and above the study-specific estimates, and thus clarify the role of androgens as biomarkers for, or causal contributors to, men's health. The work outlined in this document will be conducted from 1 February 2019 to 30 November 2020.

### Objectives

The AIMS will establish an international collaboration of existing cohort studies to clarify the relation of endogenous sex hormones with major health outcomes in men. 'Population, exposure, outcomes' characteristics include adult men in the general community; an exposure of endogenous circulating sex hormone concentration, primarily testosterone, as the principal male sex hormone or androgen; and a prospective cohort study type, with incident health outcomes including incidence of cardiovascular disease events, mortality, cancers and dementia. The specific objectives of the AIMS are to investigate associations between variables representing social, demographic and behavioural factors with the measured concentration of testosterone in the blood of men (Analysis 1); to examine the associations between testosterone concentrations and subsequent incidence of cardiovascular events, cardiovascular deaths and all-cause mortality in men (Analysis 2); to examine the associations between testosterone concentrations and subsequent mortality from (and, if available, diagnoses of) common cancers in men (Analysis 3) and to examine the associations between testosterone concentrations and cognitive impairment and incident dementia in men (Analysis 4). Analysis 2 will evaluate myocardial infarction, stroke and heart failure, deaths due to cardiovascular disease and the composite endpoint of MACE comprising non-fatal myocardial infarction, non-fatal stroke and deaths due to cardiovascular disease. Analysis 3 will evaluate outcomes of deaths due to and diagnoses of colorectal cancer, lung cancer and prostate cancer.

## METHODS AND ANALYSIS

IPD meta-analyses will be conducted to understand the associations between testosterone and a range of associated major health outcomes in men. IPD meta-analyses have been selected as the most suitable approach because (1) the required AD are not available from each of the cohorts in published literature; (2) IPD meta-analyses

typically have higher statistical power than AD meta-analyses and provide scope for controlling for important confounders and risk factors.[37 38] Where possible, methods will adhere to the guidelines for Preferred Reporting Items for Systematic Reviews and Meta-Analyses of IPD data (completed checklist is provided in online supplementary material) and the Meta-analysis Of Observational Studies in Epidemiology.[41–43]

### Studies selected for inclusion in IPD meta-analyses

Studies will be identified by two independent reviewers from a systematic review using online search tools for mainstream published (MEDLINE, EMBASE) and grey literature (OpenGrey, Mednar) studies, conducted from 14 June 2019 to 31 December 2019. Eligible studies include prospective cohort studies with plasma or serum testosterone concentrations measured using mass spectrometry with at least 5 years of follow-up data, with incident cardiovascular, cancer, mortality, dementia or cognitive events recorded (see online supplementary table S1 for an example of search criteria to be used). The search strategy selects for articles based on words or Medical Subject Headings terms matching the relevant exposure (example steps 1–3), outcomes (example steps 4–10) and study type (example steps 14–15), then, depending on the search tool, filters down to more relevant studies, with exclusions of clinical trials and of studies on non-humans (example steps 18–27), with no date range restrictions. Ahead of the systematic review, nine eligible studies (cohorts) had expressed interest to collaborate: three from Australia (Busselton Health Study,[44] Health In Men Study,[45] Men Androgen Inflammation Lifestyle Environment and Stress Study[46]); two from Europe (European Male Ageing Study,[47] Osteoporotic Fractures in Men Study, Sweden[48]) and four from the USA (Atherosclerosis Risk in Communities Study,[49] Cardiovascular Health Study,[50] The Framingham Heart Study,[51] Osteoporotic Fractures in Men Study, USA[52]). Investigators from eight of these studies have confirmed availability of suitable IPD-level data, provisional on approvals from their respective Publication and Steering Committees. If it is not possible to obtain IPD-level data from selected studies, we will request suitable AD-level data.

### Data provision, merging, harmonisation and storage

The project manager will liaise directly with the nominated contact person for each cohort study to identify, specifically, which variables and set of observations will be suitable to request. A data request will then be submitted to the data custodian for each study. Requested variables will be labelled as either 'highly desirable' or 'only if available', in order to prioritise efforts in obtaining the key variables for analyses and to acknowledge the differing availability of variables among studies. A list of variable names, definitions and attributes, including numbers of rows and columns in each data file(s) will also be requested to be provided separately. Methods for ascertaining

outcome and comorbidity status will be requested, which can be used to indicate the relative quality of diagnostic information (eg, International Classification of Diseases (ICD)-coded diagnoses from hospital inpatient admissions vs self-report information). File transfer will be achieved via encrypted file transfer (or other sufficiently secure method).

Once each dataset has been received by the project manager, the original file(s) will be saved and date-stamped in a secure central repository. All subsequent manipulations will be completed using copies of the original file, with syntax saved as script files. Variable definitions will be checked, variables inspected for missing values and variable properties and value ranges assessed to identify possible outliers. A table of summary statistics will be calculated and, where possible, analyses run and compared with published values to check for data consistency. The nature of any discrepancies identified from these checks will need to be understood, and possibly resolved, prior to proceeding with the meta-analyses.[42 53 54]

Depending on the extent of missing-ness, missing data will be suitably imputed.[54 55] For each analysis, we will conduct multiple imputations using a method that approximates as close as possible to the substantive model.[56] The method will likely vary depending on the analysis[57] and therefore we will consider adapting either a fully conditional specification[58] or joint modelling framework[59] to each case, as is appropriate. The quality of imputations will be quantified by using re-imputed values to calculate the posterior predictive p values of relevant quantities.[60] Results from each of the multiply-imputed datasets will be suitably pooled to obtain final estimates, SEs and 95% CIs.[61]

Prior to the merging of datasets, a variable to identify each source study will be appended as the new first column. Variable formats will be checked and corrected for consistency and participant identifier codes anonymised for uniqueness across all studies. Harmonisation will be required for some variables (eg, physical activity, alcohol consumption) that are recorded differently by the different component studies. Where possible, AD datasets will be used to reconstruct IPD-level data (that is, partially reconstructed IPD) prior to merging.[62–64] Should this not be possible, the IPD-level data will be aggregated and summary estimates made with and without AD-level data as sensitivity analyses. It is also possible that some studies might have outcomes available for some analyses, but not others; in these cases, we will preferentially use IPD-level data when available but also seek to use AD-level data from those other studies when available. There is no requirement to use IPD-level data from the same studies across all analyses.

All IPD-level data will be accessible to only the approved staff from a secure on-site facility, from rooms that are kept locked when unattended and with remote access not permitted. IPD-level data are not to be printed in hard copy and will only be presented at aggregate level. Data analysed for this study will be retained for 5 years after the

**Table 1** Variables planned to be included in IPD meta-analysis modelling*

| Type | Analysis 1 | Analysis 2 | Analysis 3 | Analysis 4 |
|---|---|---|---|---|
| Outcomes/DV | Androgen concentration† | Incident CVD‡ | Incident deaths (cancer‡) | Incident dementia |
| | | Incident deaths (CVD‡) | Incident cancer‡ (diagnoses) | Baseline cognition |
| | | Incident deaths (all-cause) | | Change in cognition |
| Focal predictor | – | Androgen concentration† | Androgen concentration† | Androgen concentration† |
| Covariates/IV demographic | Age | Age | Age | Age |
| | Education level | Education level | Education level | Education level |
| | Ethnicity | Ethnicity | Ethnicity | Ethnicity |
| | Marital status | Marital status | Marital status | Marital status |
| | Site | Site | Site | Site |
| Risk factors and comorbidities | Alcohol consumption | Alcohol consumption | Alcohol consumption | Alcohol consumption |
| | BMI, waist | BMI, waist | BMI, waist | BMI, waist |
| | Physical activity | Physical activity | Physical activity | Physical activity |
| | Smoking status | Smoking status | Smoking status | Smoking status |
| | BP, hypertension | BP, hypertension | BP, hypertension | BP, hypertension |
| | General health | General health | General health | General health |
| | | Atrial fibrillation | | |
| | Prevalent CVD | Prevalent CVD | | |
| | Prevalent cancer‡ | | Prevalent cancer‡ | |
| | Prevalent dementia | | | Prevalent dementia |
| | Baseline cognition | | | |
| | COPD | COPD | | |
| | Diabetes | Diabetes | Diabetes | Diabetes |
| | Cholesterol, LDL, HDL | Cholesterol, LDL, HDL | | |
| | Creatinine level | Creatinine level | | |
| | Lipid lowering medications | Lipid lowering medications | | |
| | Anxiety | | | Anxiety |
| | Depression | | | Depression |
| | Psychotropic drug use | | | Psychotropic drug use |

*Black font: highly desirable; Green font: only if available.
†Androgens: total testosterone, dihydrotestosterone, oestradiol, luteinising hormone, sex hormone binding globulin.
‡Subgroup analyses are also planned. For CVD outcomes: heart failure, myocardial infarction, stroke. For cancer outcomes: colorectal cancer, lung cancer, prostate cancer.
BMI, body mass index; BP, blood pressure; COPD, chronic obstructive pulmonary disease; CVD, cardiovascular disease; DV, dependent variable; IV, independent variables.

last of all proposed analyses are published and will then be destroyed. A post-analysis retention period is required to enable publication and possible scrutiny of findings. Disposal will be carried out according to best practice.

### Data items

The full list of generic variables to include in meta-analyses is presented in table 1. Analysis 1 will model relationships of total testosterone concentrations, as measured in serum or plasma samples (dependent variable), with key demographic variables and risk factors for disease (predictor variables). Time-to-event variables obtained from follow-up data will be analysed in Analysis 2 and 3 (outcomes), and their associations with testosterone concentrations (focal predictor) and other potential confounders and risk factors (predictor variables). Records of dementia diagnoses (physician or otherwise categorised) and relative cognitive function (for example, from test scores) will be analysed in Analysis 4, and their associations with testosterone concentrations and other potential confounders and risk factors. Analyses of relationships with other sex hormones instead of testosterone, including dihydrotestosterone, oestradiol,

luteinising hormone (LH) and sex hormone binding globulin (SHBG), will be conducted where sufficient data are available. For exploratory analyses, free testosterone, the amount in the circulation which is not protein-bound, will be calculated from measured total testosterone and SHBG.[65] Covariates were selected to include those used in previous studies, as well as those that are typically recorded. The full list of proposed variables to include in the meta-analyses is presented in table 1.

### Statistical analysis

Since the datasets to be analysed are sampled from different populations, a random-effects meta-analysis is appropriate, as it acknowledges that effects will vary among studies due to differences in local factors.[66] One-stage IPD meta-analyses will be performed. Each IPD meta-analysis will involve fitting a model with study estimated as either a fixed or random term, to account for related observations within studies, and testosterone modelled as random slopes (when modelled as a continuous predictor) or intercepts (when modelled as a categorical predictor), to harmonised, merged data from all cohorts. The underlying statistical model and estimates of effect size will be specific to each of the proposed analyses and are outlined as follows.

### Analysis 1: factors associated with testosterone concentrations in men and characterisation of reference ranges

Linear mixed models (LMMs) or generalised linear mixed models (GLMMs) will be used to model the relation between the predictors (independent variables) and each hormonal variable (testosterone, dihydrotestosterone, oestradiol, LH and SHBG) as five separate IPD meta-analyses. Suspected non-linear relationships, at the scale of the linear predictor, will be investigated and modelled appropriately (eg, splines with pre-set knot locations and linear boundary constraints). Measures of effect size may include (but are not limited to): $\eta^2$,[67 68] Pearson's r, standardised mean difference to the reference level (categorical predictors) and standardised difference for an increase in one SD (continuous predictors). In the case of non-linear relations, we will graphically describe the relationship with comparisons made with appropriate reference points. Reference ranges will be derived based on the distributions of testosterone in healthy men.

### Analysis 2: associations between testosterone concentrations and subsequent incidence of cardiovascular events, cardiovascular deaths and all-cause mortality

Cox proportional hazards models will be used to assess the effect of testosterone level on the incident risk of each outcome, with separate IPD meta-analyses conducted for each outcome (myocardial infarction, stroke, heart failure, deaths due to cardiovascular disease and MACE) and each hormonal variable (testosterone, dihydrotestosterone, oestradiol, LH and SHBG). Component study will be modelled either as a stratified variable[39] or as a random term, and testosterone as a random term, using frailty models, which are a class of survival models that incorporate random effects.[69–71] Participants with prevalent cardiovascular disease at baseline will be excluded. The length of follow-up will also be standardised among studies in order to maximise data from all datasets, while minimising the prospect for variable lengths-to-follow-up among studies introducing additional heterogeneity into results.[55]

Multivariable versions of each of these models will also be fitted, with additional predictors for potential confounders and risk factors included (table 1). Non-linear associations for continuous variables will be modelled using natural splines with pre-specified knots and linear boundary constraints. The (standardised) measure of effect size used will be the HR. Subgroup analyses will be conducted separately for each of three specific types of cardiovascular disease (outcomes): myocardial infarction, stroke and heart failure.

### Analysis 3: association between testosterone level and subsequent incidence of cancer

Cox proportional hazards models will be used to assess the effect of testosterone concentrations on the incident risk of cancer deaths and, if available, of cancer diagnoses. IPD meta-analyses will be conducted separately for each of these outcomes and for each hormonal variable as focal predictor (testosterone, dihydrotestosterone, oestradiol, LH and SHBG). Component study will be modelled either as a stratified variable[39] or as a random term, and testosterone as a random term, using frailty models.[69–71] Participants with prevalent cancer diagnosis at baseline will be excluded. The length of follow-up will be standardised among studies in order to maximise data from all datasets, while minimising the prospect for variable lengths-to-follow-up among studies introducing additional heterogeneity into results.[55]

Multivariable versions of each of these models will also be fitted, with additional predictors for potential confounders and risk factors included (table 1). Non-linear associations for continuous variables will be modelled using natural splines with pre-specified knots and linear boundary constraints. The (standardised) measure of effect size used will be the HR. Subgroup analyses will be conducted separately for each of three common types of cancers in men (outcomes): colorectal cancer, lung cancer and prostate cancer. For these analyses, men with the relevant cancer type at baseline will be excluded from that specific analysis. Thus men with prevalent colorectal cancer (but not other cancer types) will be excluded in the analysis of incident colorectal cancer; similarly for the analyses of incident lung and prostate cancer, men with lung or prostate cancer at baseline will be excluded.

### Analysis 4: associations of testosterone levels with cognitive impairment and incident dementia in men

LMMs and GLMMs will be used to model the association of testosterone concentrations with cognitive impairment (cross-sectional analyses of baseline data). Cox

proportional hazards models will be used to assess the effect of testosterone concentrations on the incident risk of dementia. Men with prevalent dementia will be excluded from this analysis. We will also ask for follow-up cognition test scores, and if available, will run an analysis of changes in cognition test scores from baseline as an outcome. Separate IPD meta-analyses will be conducted for each hormonal variable as a focal predictor (testosterone, dihydrotestosterone, oestradiol, LH and SHBG). Component study will be modelled as a fixed or random intercept term in LMMs and GLMMs and as a stratified factor or frailty model random term in Cox regressions. Multivariable versions of each of these models will also be fitted, with additional predictors for potential confounders and risk factors included (table 1). Suspected non-linear relationships, at the scale of the linear predictor, will be investigated and modelled using splines with pre-specified knots and linear boundary constraints. Measures of effect size may include: $\eta^2$, Pearson's r, standardised mean difference to the reference level (categorical predictors), standardised difference for an increase in one SD (continuous predictors), OR for LMMs and GLMMs and HR for Cox regressions.

Throughout all analyses, contour-enhanced funnel plots will be constructed to visually assess patterns in estimates of effect sizes and precision among studies, to investigate heterogeneity and possible meta-biases.[72–74] The relative amount of heterogeneity will be estimated (eg, using $I^2$) and forest plots presented. Subgroup or meta-regression analyses may be conducted if pronounced heterogeneity is estimated.[66]

### Patient and public involvement

This IPD meta-analysis will use existing secondary data. Patients and public were not involved in the design, recruitment or conduct of this IPD meta-analysis. The results of this study will be shared with the primary investigators of the shared studies and disseminated as publications in open-access journals.

### ETHICS AND DISSEMINATION

Ethics approvals obtained for each of the participating cohorts state that participants have consented to have their data collected and used for research purposes. Furthermore, there are no expected harms or risks to participants, as data have already been collected within each individual cohort, under existing ethics approvals. De-identified data will be collated and analysed, with no new procedures planned for participants. The AIMS has been assessed as exempt from ethics review by the Human Research Ethics Office at the University of Western Australia (file reference number RA/4/20/5014). Research findings will be disseminated to the scientific and broader community via the publication of the four planned research articles, at scientific conferences and meetings of relevant professional societies (including the Endocrine Society) to ensure the clinical translation and uptake of findings. Collaborating cohort studies each have their own individual policies and strategies in place for the dissemination of findings to study participants and local communities.

### DISCUSSION

Although several published meta-analyses have investigated associations of endogenous testosterone with health outcomes in men,[75–86] none have conducted IPD meta-analyses of health outcomes as planned for this study. A previous IPD meta-analysis focussed on the outcome of metabolic syndrome.[78] Results from this study will improve on previously published estimates from individual studies, in terms of the generalisability of findings. Estimates from the IPD meta-analyses are also likely to be more reliable than those published from conventional meta-analyses because they typically have higher statistical power and provide scope for controlling for important confounders and risk factors.[37 38] Uncertainty will undoubtedly remain due to the possibility of confounding influences of unadjusted effects. However, unlike a randomised controlled trial, this study avoids the need to subject individuals to interventions, and provides more comprehensive characterisation of temporal relationships between baseline testosterone concentrations and a range of key health outcomes.[87]

Accordingly, it is hoped that the AIMS collaboration will ultimately complement the research efforts and outputs from multiple prospective cohort studies by drawing on the collective body of evidence to clarify the role of endogenous sex hormone levels on major health outcomes in men. It is possible that this work might also elucidate new understanding, arising from improved scope for fitting more complex models due to increased statistical power or from patterns detected in subgroup or meta-regression analyses. Clinically, research outputs will be used to identify the scope and optimal recruitment criteria for future trials of testosterone therapy. These data will also allow reference ranges for testosterone in men across ages and geographical locations to be refined, to inform recommendations for clinical practice more generally.

**Author affiliations**
[1]Medical School, University of Western Australia, Perth, Western Australia, Australia
[2]Department of Endocrinology and Diabetes, Fiona Stanley Hospital, Perth, Western Australia, Australia
[3]School of Population and Global Health, University of Western Australia, Perth, Western Australia, Australia
[4]Adelaide Institute for Sleep Health, Flinders University, Bedford Park, South Australia, Australia
[5]Clinical and Experimental Endocrinology, KU Leuven, Leuven, Belgium
[6]Internal Medicine, Baylor College of Medicine, Houston, Texas, USA
[7]Harvard Medical School, Boston, Massachusetts, USA
[8]San Francisco Coordinating Center, California Pacific Medical Center Research Institute, San Francisco, California, USA
[9]Gillings School of Global Public Health, University of North Carolina at Chapel Hill, Chapel Hill, North Carolina, USA
[10]School of Medicine, Division of Endocrinology, Diabetes and Metabolism, Johns Hopkins University, Baltimore, Maryland, USA

[11]WA Centre for Health & Ageing, University of Western Australia, Perth, Western Australia, Australia

[12]Department of Clinical Sciences and Orthopedic Surgery, Lund University, Lund, Sweden

[13]Freemasons Foundation Centre for Men's Health, The University of Adelaide, Adelaide, South Australia, Australia

[14]Geriatric Research, Education and Clinical Center, VA Puget Sound Health Care System, Seattle, Washington, USA

[15]Department of Medicine, Division of Gerontology & Geriatric Medicine, University of Washington School of Medicine, Seattle, Washington, USA

[16]Centre for Bone and Arthritis Research at the Sahlgrenska Academy, Institute of Medicine, University of Gothenburg, Goteborg, Sweden

[17]Oregon Health & Science University, Portland, Oregon, USA

[18]Centre for Epidemiology Versus Arthritis, Faculty of Biology, Medicine and Health, The University of Manchester & NIHR Manchester Biomedical Research Centre, Manchester, UK

[19]Manchester University NHS Foundation Trust, Manchester Academic Health Science Centre, Manchester, UK

[20]VA Puget Sound Health Care System, Seattle, Washington, USA

[21]School of Medicine, Department of Psychiatry and Behavioral Sciences, University of Washington, Seattle, Washington, USA

[22]Institute for Aging Research, Hebrew SeniorLife, Beth Israel Deaconess Medical Center, Boston, Massachusetts, USA

[23]Department of Chronic Diseases, Metabolism and Ageing (CHROMETA), Laboratory of Clinical and Experimental Endocrinology, Katholieke Universiteit Leuven, Leuven, Flanders, Belgium

[24]Division of Diabetes, Endocrinology and Gastroenterology, The University of Manchester, Manchester, UK

**Acknowledgements** We thank Östen Ljunggren of the Department of Medical Sciences, Uppsala University and Liesbeth Vandenput of the Centre for Bone and Arthritis Research at the Sahlgrenska Academy Institute of Medicine, University of Gothenburg, Mary Lou Biggs, Terena Solomons, Jos Tournoy, Mark Divitni, Matthew Knuiman, John Acres, the staff and participants of the ARIC Study, the Busselton Population Medical Research Foundation for access to the data, the Cardiovascular Health Study, the EMAS Study, the Health In Men Study, the MAILES Study, the MrOS Study, the Framingham Heart Study for their important contributions.

**Collaborators** The Androgens In Men Study: (1) Busselton Health Study: Busselton Population Medical Research Institute (Inc.), Department of Pulmonary Physiology and Sleep Medicine, Sir Charles Gairdner Hospital, Nedlands WA 6009, Australia (2) Health In Men Study (HIMS): Western Australian Centre for Health and Ageing, University of Western Australia, Level 6 Medical Research Foundation Building, Royal Perth Hospital, Rear 50 Murray Street, Perth, Western Australia 6000, Australia. (3) Men Androgen Inflammation Lifestyle Environment and Stress (MAILES) Study: Freemasons Foundation Centre for Men's Health, University of Adelaide, 254 North Tce, Adelaide, Australia. (4) European Male Ageing Study (EMAS): The University of Manchester, Stopford Building, Oxford Road, Manchester, M13 9PT, UK. (5) Osteoporotic Fractures in Men (MrOS) Study Sweden: Department of Clinical Sciences and Orthopedic Surgery, Lund University, Skåne University Hospital, Malmö, Sweden; MrOS Study USA: California Pacific Medical Center Research Institute, San Francisco Coordinating Center, 550 16th street, San Francisco, CA 94143. (6) Atherosclerosis Risk in Communities Study (ARIC): School of Public Health, U. of North Carolina, Suite 2013, NCNB Plaza, 137 E. Franklin St, Chapel Hill, NC 27514. (7) Cardiovascular Health Study: National Heart, Lung, and Blood Institute, Bethesda, MD. (8) The Framingham Heart Study, Boston University School of Medicine, 72 E. Concord Street, B-601, Boston, MA 02118.

**Contributors** BBY, KM, LA, SB, ASD, AMM, CO, ESO, TWO, DV and GAW contributed to the study concept and design. BBY, KM and RJM prepared the initial draft of the manuscript, including literature review, list of variables and statistical methods. All the authors were involved in subsequent revisions to the protocol and manuscript, and approved this submission.

**Funding** AIMS is funded by a Western Australian Health Translation Network (WAHTN) Medical Research Future Fund Rapid Applied Translation Grant (2018). The ARIC study is funded by has been funded in whole or in part with Federal funds from the National Heart, Lung, and Blood Institute, National Institutes of Health, Department of Health and Human Services, under Contract nos. (HHSN268201700001I, HHSN268201700002I, HHSN268201700003I, HHSN268201700005I, HHSN268201700004I) and Dr. Ballantyne received NIH grant support through R01HL134320 The 1994/1995 Busselton Health Survey was supported by the Western Australian Health Promotion Foundation. The Cardiovascular Health Study was supported by contracts HHSN268201200036C, HHSN268200800007C, HHSN268201800001C, N01HC55222, N01HC85079, N01HC85080, N01HC85081, N01HC85082, N01HC85083, N01HC85086, and grants U01HL080295 and U01HL130114 from the National Heart, Lung, and Blood Institute (NHLBI), with additional contribution from the National Institute of Neurological Disorders and Stroke (NINDS). Additional support was provided by R01AG023629 from the National Institute on Aging (NIA). A full list of principal CHS investigators and institutions can be found at CHS-NHLBI.org. The EMAS Study is funded by The Commission of the European Communities Fifth Framework Programme 'Quality of Life and Management of Living Resources' Grant No. QLK6-CT-2001-00 258 and supported by Arthritis Research UK. The Framingham Heart Study of the National Heart Lung and Blood Institute of the National Institutes of Health and Boston University School of Medicine has been funded in whole or in part with Federal funds from the National Heart, Lung, and Blood Institute, National Institutes of Health, Department of Health and Human Services, under Contract No. 75N92019D00031. The HIMS study is funded by project grants from the National Health and Medical Research Council of Australia. The MAILES Study is funded by a project grant from the National Health and Medical Research Council of Australia (627227). The MrOS Sweden Study is funded by The Swedish Research Council and Swedish ALF funding. The Osteoporotic Fractures in Men (MrOS) Study is supported by National Institutes of Health funding. The following institutes provide support: the National Institute on Aging (NIA), the National Institute of Arthritis and Musculoskeletal and Skin Diseases (NIAMS), the National Center for Advancing Translational Sciences (NCATS) and NIH Roadmap for Medical Research under the following grant numbers: U01 AG027810, U01 AG042124, U01 AG042139, U01 AG042140, U01 AG042143, U01 AG042145, U01 AG042168, U01 AR066160 and UL1 TR000128.

**Disclaimer** The funders played no role in the study design, data collection, data analysis or interpretation, writing of the manuscript or the decision to submit the article for publication. All authors are accountable for all aspects of the work in ensuring that questions related to the accuracy or integrity of any part of the work are appropriately investigated and resolved. The content is solely the responsibility of the authors and does not necessarily represent the official views of the National Institutes of Health.

**Competing interests** None declared.

**Patient and public involvement** Patients and/or the public were not involved in the design, or conduct, or reporting, or dissemination plans of this research.

**Patient consent for publication** Not required.

**Provenance and peer review** Not commissioned; externally peer reviewed.

**ORCID iDs**
Ross James Marriott http://orcid.org/0000-0002-8805-8498
Kevin Murray http://orcid.org/0000-0002-8856-6046

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
