## [Reviewer comments · BMJ Open]

ARTICLE DETAILS

TITLE (PROVISIONAL)	Androgens In Men Study (AIMS): protocol for meta-analyses of individual participant data investigating associations of androgens with health outcomes in men.
AUTHORS	Yeap, Bu; Marriott, Ross; Adams, Robert; Antonio, Leen; Ballantyne, Christie; Bhasin, Shalender; Cawthon, PM; Couper, David; Dobs, Adrian; Flicker, Leon; Karlsson, Magnus; Martin, Sean A.; Matsumoto, AM; Mellström, Dan; Norman, Paul; Ohlsson, C; Orwoll, Eric; O'Neill, Terence; Shores, Molly; Travison, Thomas; Vanderschueren, Dirk; Wittert, Gary; Wu, Frederick; Murray, Kevin

VERSION 1 – REVIEW

REVIEWER	Benjamin Hsu UNSW, Centre for Big Data Research in Health
REVIEW RETURNED	31-Oct-2019

GENERAL COMMENTS	Well planned piece of research. This is a worthwhile attempt to address the challenging question on the causal relationship between various sex hormones and these poor health outcomes. It may also be valuable using this dataset to explore the relationship between sex hormones with alterations in bone health, body compositions and physical functions. Many of the included cohort studies have sex hormone measurements at multiple study waves. The authors should take advantage of this since age-related changes in testosterone or other sex hormones may significantly contribute to the development of those identified poor health outcomes in older men.
--

REVIEWER	Dr Patrick J Owen Deakin University, Institute for Physical Activity and Nutrition
REVIEW RETURNED	11-Nov-2019

GENERAL COMMENTS	The authors present an interesting study concept in which data from several cohorts will be pooled for analyses. The study is strengthened by the unique approach and thorough statistical analyses. I wish the authors all the best with this ambitious and clinically important research. Notably, this was written and formatted to an exceptionally high standard. Specific comments are as follows: MAJOR
--

	1. Ensure in the final version that a detailed summary of the search strategy and hits associated are presented and available as supplementary data. This will aid in replication of the systematic review in the future, as well as serve as a learning tool for those looking to design and implement their own reviews with similar MeSH etc. terms 2. Page 11, Line 227: 'Missing data will be suitably imputed' whilst I appreciate references are provided, greater detail of what will be done is warranted, as this approach should be established now, rather than once data is collected/analysed. Please include this in the protocol manuscript. 3. At times, testosterone and androgen conception appear to be used interchangeably. This should be updated throughout to specifically detail what the variable examined will be. Pooling together androgens will impact outcomes given variance in concentration. Moreover, considering whether serum/plasma variables can be pooled should also be justified. As for other 'generic variables', I appreciate this is as much detail as you can provide given the progress of the study. MINOR 1. Page 4, Line 76: 'The date range of bibliographic searches will not be constrained, and aggregate level data will be sought where individual participant data (IPD) is not available' is confusing. 2. Page 4, Line 85: 'There are four planned research articles, with each involving a separate set of IPD meta-analyses' greater detail could be provided for this statement, even if just in brief re: differences in these papers 3. Introduction, Para 1: Given age is clearly an important factor re: T levels, it would be ideal to quantify what is meant by 'older' and 'young'. Similarly, able to quantify an average decline per year/decade of ageing, this would be helpful for interpreting the effects of ageing. 4. Introduction, Para 2: Defining ranges deemed 'hypogonadal' or 'low-normal' is warranted. These values tend to vary; thus, it should be clear what is meant by these terms. 5. Page 15, Line 320: 'frailty models' should be elaborated upon and the rationale for inclusion within the analyses is warranted. 6. Page 15, Line 320: It may be worthwhile including those with cancer Dx at baseline to examine recurrence. Given you will have these data, it could be another outcome worth considering. 7. Analysis 3: Why only colorectal, lung and prostate only re: sub-group analyses? I would assume data on other types of cancer will be available, with analyses warranted with noted limitations if sample sizes are small.
--	--

VERSION 1 – AUTHOR RESPONSE

Response: The Conclusions section has been deleted.

2. Reviewer(s)' Comments to Author:

Reviewer: 1

Reviewer Name: Benjamin Hsu

Institution and Country: UNSW Sydney, Australia Please state any competing interests or state 'None declared': None declared

Please leave your comments for the authors below Well planned piece of research. This is a worthwhile attempt to address the challenging question on the causal relationship between various sex hormones and these poor health outcomes.

Response: Thank you.

It may also be valuable using this dataset to explore the relationship between sex hormones with alterations in bone health, body compositions and physical functions.

Response: Thank you for the suggestions. This protocol article describes only the analyses to be done on cardiovascular, cancer, cognitive decline and dementia outcomes, as budgeted, and as outlined in the plan submitted to PROSPERO. The authors have considered investigating associations of androgens with other health outcomes, including fractures, but these would be the subject of a new proposal.

Many of the included cohort studies have sex hormone measurements at multiple study waves. The authors should take advantage of this since age-related changes in testosterone or other sex hormones may significantly contribute to the development of those identified poor health outcomes in older men.

Response: This is a good suggestion, however we know that not all of the component studies will have testosterone measured at multiple time points, so cannot include this feature in the one-stage IPD meta-analysis modelling. The planned pooled analyses will be limited to include only those variables that are common to all component studies. In the future we may revisit this and include longitudinal androgen measurements in the analyses where available, in a new proposal.

Reviewer: 2

Reviewer Name: Dr Patrick J Owen

Institution and Country: Deakin University, School of Exercise and Nutrition Sciences, Geelong, Victoria, 3220, Australia Please state any competing interests or state 'None declared': N/A

Please leave your comments for the authors below The authors present an interesting study concept in which data from several cohorts will be pooled for analyses. The study is strengthened by the unique approach and thorough statistical analyses. I wish the authors all the best with this ambitious and clinically important research. Notably, this was written and formatted to an exceptionally high standard.

Response: Thank you.

Specific comments are as follows:

MAJOR

1. Ensure in the final version that a detailed summary of the search strategy and hits associated are presented and available as supplementary data. This will aid in replication of the systematic review in the future, as well as serve as a learning tool for those looking to design and implement their own reviews with similar MeSH etc. terms

Response: The MEDLINE search strategy is presented as supplementary data (i.e., Supplementary Table S1). This includes MeSH terms as entered into MEDLINE. Bearing in mind this is a protocol paper, once the systematic review has been completed, that publication will include more specific

details on the search strategies and hits (i.e., results) for each of the search tools used for both the published (MEDLINE, EMBASE) and grey (OpenGrey, Mednar) literature items. A brief statement in the Supplementary Material document has been added to make this point, accordingly.

2. Page 11, Line 227: 'Missing data will be suitably imputed' whilst I appreciate references are provided, greater detail of what will be done is warranted, as this approach should be established now, rather than once data is collected/analysed. Please include this in the protocol manuscript.

Response: The development of imputation methods for IPD meta-analyses is an active area of research, and there is currently no one commonly-accepted approach suitable in all cases. However, there are some aspects to imputation for these types of analyses that are gaining support, including:

- i. Multiple imputations for minimising the bias in estimates and standard errors of estimates;
- ii. Using an imputation model that is appropriate to the assumed missing data model and substantive analysis model; and
- iii. Ensuring that the grouping/cluster structure of mixed effects models, and ideally the heterogeneity in error variances among clusters (if present), is preserved in the imputed values.

We have inserted additional text to address these points with our plan for imputing missing data, and note that the approach will likely differ with each of the analyses planned, accordingly. We also have outlined a strategy for quantifying and reporting the quality of imputations made in each analysis. (Page 11, Lines 232-240)

3. At times, testosterone and androgen conception appear to be used interchangeably. This should be updated throughout to specifically detail what the variable examined will be. Pooling together androgens will impact outcomes given variance in concentration. Moreover, considering whether serum/plasma variables can be pooled should also be justified. As for other 'generic variables', I appreciate this is as much detail as you can provide given the progress of the study.

Response: We have clarified in the Introduction that we are referring to sex hormones ("comprising testosterone and dihydrotestosterone as the major androgens, and estradiol as the major estrogen, in the circulation" page 7, lines 144-145) and in the remainder of the text we have specified when the independent variable is testosterone, DHT or estradiol. Mass Spectrometry provides accurate measurement of T, DHT, E2 using either serum or plasma. (e.g., Star-Weinstock & Dey, 2019)

Reference cited:

Star-Weinstock M, Dey S. 2019. Development of a CDC-certified total testosterone assay for adult and pediatric samples using LC-MS/MS. *Clinical Mass Spectrometry* 13: 27-35.

MINOR

1. Page 4, Line 76: 'The date range of bibliographic searches will not be constrained, and aggregate level data will be sought where individual participant data (IPD) is not available' is confusing.

Response: This sentence has been revised and split into 2 sentences for clarity. No date range was set for the searches.

2. Page 4, Line 85: 'There are four planned research articles, with each involving a separate set of IPD meta-analyses' greater detail could be provided for this statement, even if just in brief re: differences in these papers

Response: "(articles will investigate different, distinct outcomes)" added.

3. Introduction, Para 1: Given age is clearly an important factor re: T levels, it would be ideal to quantify what is meant by 'older' and 'young'. Similarly, able to quantify an average decline per year/decade of ageing, this would be helpful for interpreting the effects of ageing.

Response: On page 6, line 123 we replaced the word "older" with more specific text for that cited research as "aged 65 and older". The primary aim of this work as outlined in the protocol is to assess testosterone in relation to these outcomes and adjust for confounders, including age. Not to look at the effects of age in particular. However, if the data warrant, we may perform sensitivity analyses looking at associations in men < 40, 40-70 and 70 and above.

4. Introduction, Para 2: Defining ranges deemed 'hypogonadal' or 'low-normal' is warranted. These values tend to vary; thus, it should be clear what is meant by these terms.

Response: These definitions are contained within the cited references. See 26-29 and 30-34 respectively.

5. Page 15, Line 320: 'frailty models' should be elaborated upon and the rationale for inclusion within the analyses is warranted.

Response: Thank you, but we believe that we have already satisfactorily covered this point in text:

- i. In the previous section for Analysis 2 we stated that frailty models "are a class of survival models that incorporate random effects" (page 14, lines 314-315);
- ii. Within each subsection of the "Statistical analysis" section in the Methods, where we described the specific details of each of the respective planned analyses, we also clearly stated what the random effects to be modelled were; and
- iii. Modelling the effect of interest as a random term is consistent with a random-effects meta-analysis, which we have stated on page 4, line 77, but also on page 13, lines 283-285 where we clearly state the rationale for performing random-effects (as opposed to fixed-effects) meta-analyses.

This specification for the planned analyses using frailty models is consistent with statistical formulations provided for one-stage random-effects IPD meta-analyses by Burke et al. (2016).

Reference cited:

Burke DL, Ensor J, Riley RD. Meta-analysis using individual participant data: one-stage and two-stage approaches, and why they may differ. *Statist Med* 2016; 36:855-75

6. Page 15, Line 320: It may be worthwhile including those with cancer Dx at baseline to examine recurrence. Given you will have these data, it could be another outcome worth considering.

Response: Thank you for the suggestion, but that would be a different set of analyses to what is planned and budgeted for in this study. We will consider this idea for possible future research.

7. Analysis 3: Why only colorectal, lung and prostate only re: sub-group analyses? I would assume data on other types of cancer will be available, with analyses warranted with noted limitations if sample sizes are small.

Response: These are the major cancer types affecting men of the anticipated age of the study cohorts and other types of cancer would be outside the scope of the current proposal.

VERSION 2 – REVIEW

REVIEWER	Dr Patrick J Owen Deakin University, Institute for Physical Activity and Nutrition
REVIEW RETURNED	01-Dec-2019
GENERAL COMMENTS	I am satisfied with the authors responses - well done on addressing these to a high standard. I wish the authors all the best conducting this important research.

VERSION 2 – AUTHOR RESPONSE

Reviewer(s)' Comments to Author:

Reviewer: 2

Reviewer Name: Dr Patrick J Owen

Institution and Country:

Deakin University, School of Exercise and Nutrition Sciences, 221 Burwood Highway, Burwood, Victoria, 3125, Australia Please state any competing interests or state 'None declared': Nil

Please leave your comments for the authors below

I am satisfied with the authors responses - well done on addressing these to a high standard. I wish the authors all the best conducting this important research.

Response: Thank you.